# A Multidisciplinary Review of the Inka Imperial Resettlement Policy and Implications for Future Investigations

**DOI:** 10.3390/genes12020215

**Published:** 2021-02-02

**Authors:** Roberta Davidson, Lars Fehren-Schmitz, Bastien Llamas

**Affiliations:** 1Australian Centre for Ancient DNA, School of Biological Sciences and The Environment Institute, Adelaide University, Adelaide, SA 5005, Australia; 2UCSC Paleogenomics, University of California Santa Cruz, Santa Cruz, CA 95064, USA; lfehrens@ucsc.edu; 3UCSC Genomics Institute, University of California Santa Cruz, Santa Cruz, CA 95064, USA; 4Centre of Excellence for Australian Biodiversity and Heritage (CABAH), University of Adelaide, Adelaide, SA 5005, Australia; 5National Centre for Indigenous Genomics (NCIG), Australian National University, Canberra, ACT 0200, Australia

**Keywords:** Inka, interdisciplinary, ethnohistory, isotopes, paleogenetics, resettlement

## Abstract

The rulers of the Inka empire conquered approximately 2 million km^2^ of the South American Andes in just under 100 years from 1438–1533 CE. Inside the empire, the elite conducted a systematic resettlement of the many Indigenous peoples in the Andes that had been rapidly colonised. The nature of this resettlement phenomenon is recorded within the Spanish colonial ethnohistorical record. Here we have broadly characterised the resettlement policy, despite the often incomplete and conflicting details in the descriptions. We then review research from multiple disciplines that investigate the empirical reality of the Inka resettlement policy, including stable isotope analysis, intentional cranial deformation morphology, ceramic artefact chemical analyses and genetics. Further, we discuss the benefits and limitations of each discipline for investigating the resettlement policy and emphasise their collective value in an interdisciplinary characterisation of the resettlement policy.

## 1. Introduction

The Inka (also known as Inca) empire (1438–1533 CE) was the largest state that ever arose in pre-Columbian South America. It encompassed at its peak ~2 million km^2^ of the Andes and included deserts, mountains and coastal biomes [1] (Figure 1). Known by its people as “*Tawantinsuyu*”, approximately meaning “realm of four parts” (*Tawa* = four, *suyu* = region), the imperial state subjugated numerous pre-existing ethnic groups. The Inka monarch, or *Sapa Inka* (*Sapa* = sole) was supported by an aristocracy of related royal family groups (royal ayllus) [2]. The administrative, political and military centre of the empire was located in the city of Cusco [3] (Figure 1). In Tawantinsuyu, the pre-existing Indigenous groups were rearranged into provinces of 20,000 households, whereby larger groups were broken down and smaller groups merged with others [2]. Each province was structured into a tiered decimal hierarchy system, where taxpayers were grouped into groups from 10–10,000, with multiples of 5 or 10, such that the hierarchy of groups contained 10,000 households at the highest tier, then iteratively nested within that were groupings of 5000, 500, 100, 20 and 10 taxpayers at the lowest tier [2]. Each tier in the hierarchy was appointed an individual leader [2]. The high-level leaders would accept their role to serve the Inka, while lower-tier leaders were pre-existing respected individuals within their own communities [2]. Taxes were extracted from the heads of household, usually in the form of labour [2], though tributes of artisanal or agricultural produce were also paid to the state. Taxes were recorded with the knotted string system known as the *quipu* [4]. Although no written language was ever developed, the Inkas’ advanced engineering skills are evidenced by a 40,000-km network of roads (*Qhapaq Ñan*) [5] and megalithic architectural sites such as Machu Picchu, built to withstand earthquakes [6].

The term “Inka” has acquired a multifaceted definition. Initially, it referred to the founding Inka royal family and their ethnic descendants and relatives [7]. As they exerted political control in the Cusco region and beyond, Inka became synonymous with any people actioning the imperial expansion [7]. Then the term expanded to any individual or group in a political position that promoted or enforced the Inka doctrine, irrespective of their status as ethnic Inkas or not [7]. Later on, Indigenous tribes who natively spoke the Quechua language were elevated to a status of “Inkas by privilege” by the monarch Pachacuti, as opposed to the original ethnic group of “blood Inkas” [8]. The lines were blurred even more through the lens of foreign scholarship that treated all Indigenous Andeans collectively as one ethnicity, and the colonial destruction of separate Indigenous identities. By colonial times, “Inka” referred to any and all people of The Inka Empire/Tawantinsuyu and their achievements, resulting in a homogenised national identity [7].

Throughout the imperial expansion, the Inkas did not form the same kind of relationship with each group they encountered [9]. Rather, imperial action was customised to the political organisation, identity and ecological environment of the subjugated group [1]. Archaeological and historical evidence suggests that the initial, gradual expansion of control over the Cusco region ranged from alliance-building to intimidation and even warfare [1]. As the empire expanded, the need to maintain control over a geographically vast area prompted the Inkas to enact a series of policies, whereby (according to historical accounts) a significant fraction of the population was resettled over large distances. These policies are hereafter bundled and referred to as the resettlement or relocation policy. There is some suggestion that some elements of the resettlement policy may have had roots earlier, during the Middle Horizon Wari or Tiwanaku polities (500–1000 CE), in the form of large-scale, long-range mobilisation of workers to obtain resources not locally available [10,11,12]. However, with respect to the Inka empire, relocation would allow social and cultural manipulation of the people, rescinding the need for active military control. More specific elements of the policy contributed to the social construction of the ubiquitous “Inka” identity that would come to refer to all people indigenous to the Andes [7].

Specifically, this review sets out to answer several, intertwined questions.

What do we know about the Inka resettlement policy from the ethnohistorical record?What scientific disciplines are available that could research the reality of the resettlement policy (including magnitude, scale, impact on culture and demography, etc.) and compare it to the portrayal given by ethnohistorical documents?What contribution have scientific disciplines already made to the factual investigation and our understanding of this policy and what are the relevant benefits and limitations of those disciplines?What could future studies do to investigate the resettlement policy during the Inka empire, and what would be the best way to go about it?

## 2. Cautionary Note about Ethnohistorical Sources

There are thousands of surviving pieces of documentation that describe life under the Inkas, although only about fifty describe Inka history [10]. D’Altroy divides the sources by author type into the following categories: major 16th-century Spanish authors, mixed Andean ancestry authors with a perspective of “Christians with a foot in both cultures”, later Spanish chroniclers and Spanish colonial records from the church or court [10]. It is therefore important to consider the subjectivity of the author inherent to their identity and place in society, which would have greatly affected the inflection of their portrayal. The earliest accounts were written voluntarily by observant Spanish soldiers during the military invasion of a foreign land [10]. They were not trained anthropologists or linguists and were surrounded by people jarringly different than their own [10]. Additionally, early reports were often composed of knowledge from informants passed to the educated scribe, meaning the writer was not the same person as the observer in many cases and inaccuracies and embellishments must be expected. Modern scholars tend to place a lot of weight on these early accounts given they are the earliest source of written records. However, because the scientific method was still in its infancy in Europe in the late 16th–early 17th century, we should be mindful not to hold these accounts to the same standard as the work of contemporary linguists, anthropologists and historians. Therefore, all sources from the first colonial century must be viewed carefully and consider the authors’ motivations to write the way they did, as well as the specific historical context at the time of writing [7].

## 3. Resettlement Policy

### 3.1. Defining Terms

The peoples that were relocated under Inka rule were divided into purposeful categories for their resettlement (here the reader is invited to refer to Table A1 for alternate spellings and complete definitions of indigenous terminology used throughout the text, denoted by italics). *Aqllakuna* were chosen women (*aqllay* = “to choose” and -*kuna*, plural suffix) [12]. Selected at a young age, *aqllakuna* were removed from their family unit, housed and trained to weave and perform religious and ceremonial duties, predominantly for the rulers in Cusco [13]. Approximately the male equivalents of *aqllakuna*, the *yanakuna* (*yana* = black, dark, obscure) were similarly separated from their homes at a young age and permanently assigned to state or aristocratic service. Their role was not inherited and might have been used as a punishment mechanism, although *yana* individuals could attain promotion to higher serving positions [2]. Though not exactly synonymous with slavery, *yanakuna* have been compared to medieval European serfs [12]. A *kuraka* was the superior or elder figure in each region or province, one who knew the history of his people. The role was often assigned to existing chiefs and leaders of Indigenous ethnic groups at the time of Inka conquest [8]. *Mitmaqkuna* (*mitima* = outsider, foreigner, newcomer, one who has been relocated) were groups of people permanently resettled from one province to another for purposes of political control and cultural homogenisation [14]. The *mitmaqkuna* were released from command of their home *kuraca* and reassigned to their new provincial *kuraka* [14]. *Mitmaqkuna* were often confused by the Spaniards with *mitayoq* (*mit’a* = public labour tribute and -*yoq* = with, having, possessing), who were temporary workers doing state-mandated tribute labour in distant provinces but still a member of their home province and under command of their original *kuraka* [14]. Both *mitmaqkuna* and *mitayoq* could be employed as specialist farmers, herders, miners or artisans, producing resources or manufacturing products for the state, religious or aristocratic use [10].

### 3.2. The magnitude of Resettlement

Estimates of human population in the Andean region at first European contact range from 3.0–33.8 million [15]. As challenging as it is to estimate population size around the contact period, it is much harder to estimate the population size a century earlier, when the Inka imperial state was forming. However, there is nothing to suggest a change in population during the Inka period that would be significantly larger than the variation in population estimates. Therefore, these estimates still provide adequate context for understanding the magnitude of the relocation policy.

There is no definite number of how many people were resettled under Inka rule across Tawantinsuyu. Although the Inka kept censuses, modern scholars are still unable to decipher the quipus that record this information. However, there are several excerpts among the historical records that can give an indication. Cobo states that approximately 6000–7000 families were removed from each “province” as it was conquered, and replaced with a similar number of families removed from other provinces so that the provincial population remained constant [14]. D’Altroy estimates a total forced relocation of 3 million individuals over great distances [16]. Elsewhere, it is suggested that 14,000 foreign individuals from as far away as Chile worked the state fields at Cochabamba in central Bolivia [17], and at least 22 groups of *mitmaqkuna* were reportedly removed from the Chachapoyas region in northern Peru [18]. The accuracy of these numbers is debatable, but the purpose here is to exhibit figures approximating reality. Cobo’s account faces scepticism from later scholars because it cannot be known exactly what the total population of the empire was, and terms such as “province” and “group” are both variable and undefined. However, scholars agree with Cobo’s claim that between a quarter and a third of the total Andean population was relocated [2,10].

### 3.3. Characterising the Resettlement Policy According to Ethnohistorical Records

The resettlement policy can be reduced to the pursuit of a set of goals [2]. The first was to prevent rebellion in an empire too geographically large to be under military control [2]. This was achieved by breaking up local community groups that may have been particularly likely to rebel and systematically relocating them to distant regions. Often they were resettled in recently conquered towns where the Indigenous peoples spoke a different language, to prevent the uprising of a coherent coalition [11]. The *mitmaqkuna* were installed as the elite upper class (*hanan*) within their new communities, where the remaining Indigenous group became the lower class (*hurin*) [9]. The foreigners had the necessary political power and knowledge to dispense traditional Inka ceremonies and imperial order to the Indigenous people of that province, inevitably establishing a social disparity between the *mitmaqkuna* and the Natives [9]. Acuto contends this disparity was one of great inequality, approximating a cultural apartheid–based on archaeological surveys in north-western Argentina, where a typical pattern appears of Inka settlements close but separate from Indigenous settlements. However, Cobo restricts the inequality to the seating place and speaking order at court [14]. In any case, both authors concur that the resettlement system created a healthy conflict (from the imperial perspective) that would focus the people’s attention on local competition, which served to increase working productivity and allowed imperial observation of work capability and diligence for later recruitment [14]. Importantly, the people would be distracted from unifying into greater rebellion against the Inka state [9,14].

Another imperial goal was the construction of a productive, homogenised and coherent empire by systematic social engineering [2]. People from more populous areas were moved to less densely populated areas in an attempt to more evenly distribute the population [14]. Although it is not known to what extent, it seems that people were relocated in a climate similar to that which they were born in so they (a) would not feel adversely displaced, homesick or fall ill from the change in environment that could occur when moving from highlands to the coast or vice versa, and (b) would already have the ability to farm the land for their sustenance, as they would be familiar with the climate to grow the same crops [14].

Overall, the *mitmaqkuna* groups were effectively appointed to the role of colonist in relation to the Indigenous peoples of the region they were resettled to. Therefore, in effect, the Inka state utilised the people it conquered as tools to colonise other people it conquered, simultaneously fracturing Indigenous communities and cultures, and contributing to the ubiquitous formation of the “Inka” empire. In addition to the vast systematic displacement of people, the Inka actively built a social structure in the capital of Cusco, where colonists from each of the four parts of the empire were resettled in 12 neighbourhoods surrounding the imperial core in a manner that reflected their spatial positioning in Tawantinsuyu, creating an “ethnic microcosm” within the city [10]. Cobo suggests this contributed to maintaining power over conquered peoples. The state would translocate the main “idol” of a culture to Cusco with enough Indigenous peoples to maintain the same services/cult as in the home place. Therefore, the population of Cusco contained peoples of all conquered tribes/populations [14].

## 4. Use of Bioarchaeology to Examine the Inka Resettlement Policy

### 4.1. Isotopic Analysis

Stable isotope analysis has emerged as a tool to infer the geographic origins of buried individuals found in an archaeological site. Analysis of isotopic ratios of five elements (carbon, nitrogen, oxygen, strontium and lead) can be used to reconstruct information about the types of food consumed, climate and region of birth. The ability to compare region of birth to region of burial is particularly useful with respect to the Inka period given the reported magnitude and scale of population displacements [12].

Measuring isotopic ratios in different bodily tissues is known to reflect different stages of an individual’s life. Dental enamel reflects the first decade of life because enamel absorbs the chemical composition of foods eaten during tooth formation [19,20]. Bone tissue is continually remodelled and records an average over the last decade or two of life [21,22]. Hair keratin reflects the final few months of life, relative to hair length [23,24].

Specifically, isotopic ratios of strontium and lead are a proxy for the bedrock geology in different regions, which can be used to extrapolate geographical location against a reference map [25,26]. Oxygen isotopic ratios are influenced by the water ingested either from food, atmospheric moisture or drinking water [26]. These oxygen isotope ratios in water vary spatially and can be confounded if some of the consumed water originally fell upstream in a river catchment area [13,26].

Carbon and nitrogen are indicative of dietary composition. Given their prevalence in lipids, carbohydrates and proteins, the isotopic ratios can reflect macronutrient dietary proportions. More important than reconstructing ancient diets is the comparison of isotope ratios from enamel and bone tissue to see if a significant shift occurred from early to late life. Indeed, a major change in dietary staples could indicate migration from one region to another [27].

Isotope analysis reveals information about the geographic origins of individuals, and in studies of entire buried populations, can help to infer their social class and therefore the status-mediated demography of the site of interest, based on the underlying assumption outlined in Figure 2 [13,26,27]. If buried individuals are of local origin, they should all show the same isotopic ratio signature with no significant change from early to later life [13,26,27]. In the case of *aqllakuna* or *yanakuna*, a wide distribution with no particular modality is expected as individuals would have come from a range of different locations [13,26,27]. Conversely, for *mitmaqkuna* colonists, an isotopic variability range that has some bi- or tri-modality is expected, depending on the number of source populations of *mitmaqkuna* at a particular site [13,26,27]. This depends on the assumption from historical texts that *mitmaqkuna* were resettled wholesale, as groups extracted from a pre-existing population, while the *aqllakuna* and *yanakuna* were selected as individuals at a young age and redistributed during their lifetimes, possibly repeatedly.

These methods have been employed to study Inka period burials at several sites to infer people’s migratory origin. At Machu Picchu, researchers found that the immigrant population (*n* = 71) was largely *aqllakuna* and *yanakuna*, not *mitmaqkuna* [27]. Isotopic study of burials at Chokepukio (*n* = 59) found strong evidence for migration during the Inka period, in particular a presence of predominantly female migrants of diverse geographic origin [25]. A study of three burial locations at Túcume in north Peru offered a unique opportunity to superimpose migration and social structure. At *Cemetery Sur*, individuals were low-mobility commoners (*n* = 8), more low to mid-level-mobility elite individuals and their entourage were at *Huaca Larga* (*n* = 22) and high-mobility foreigners were at *Templo de la Piedra Sagrada* (*n* = 20) [13].

Further, comparisons between bone tissue (indicative of the last decade of life) and hair keratin (indicative of the last few months) have been used to assess changes in diet (if any) before death in Inka sacrificial victims at Chotuna-Huaca de los Sacrificios, Lambayeque, Peru (*n* = 13) [28]. Results suggested the sacrificial victims were locally born and were not fed a special diet in preparation for sacrifice [28].

There are limitations to using isotopic data by itself, such as in cases where overlap between groups can make migrant categorisation on an individual level difficult [13]. Furthermore, strontium isotopic ratios may give false positive “foreign” readings, as a result of consuming food from elsewhere so must be interpreted carefully and preferably in tandem with other forms of evidence [29]. In general, many factors can influence the tissue isotopic ratios, so measuring isotopic ratios of multiple elements allows more variation to be captured, therefore giving more power to statistical categorisations of groups and individuals.

### 4.2. Intentional Cranial Deformation Morphology

The practice of modifying the shape of the human skull has been broadly documented among pre-Columbian South American and Mesoamerican cultures [30]. Different types of intentional cranial deformation (ICD) can potentially identify natal origins since ICD is performed at a very young age and cannot be altered after the skull bones fuse [10], or if the practice is maintained by a migrant group it is still a signature of ethnicity. Although modified skulls have attracted much scholarly intrigue, there is an absence of a standardised approach to morphological characterisation and classification of the skulls, making the existing record confusing and difficult to cross-compare [31]. Recent technological developments in 3D imaging allow better measurement of crania that can differentiate between modified and unmodified specimens, e.g., [32], and also quantify standards of regularity of the practice within and between groups [31].

A modern re-analysis of the 1912 Yale Peruvian Scientific Expedition’s collection of skeletal remains from Machu Picchu addresses ICD during the Inka empire. Cranio-morphological measurements and statistical analysis support the notion of an ethnically mixed retainer population (*aqllakuna* and *yanakuna* servants) at Machu Picchu [33]. Of the sample of 50 intact skulls, 14 were of a type known as annular deformation common to various prehistoric highland groups. Another 13 had occipital flattening, a distinct type of cranial modification more typical of coastal groups, and therefore at odds with their location at the highlands site of Machu Picchu [33]. Subsequent canonical discriminant analysis using 15 measurements resulted in the classification of the Machu Picchu skulls into regions of origin when compared to reference skulls [33]. The study concluded that the typology of cranial deformations and statistical discriminant analysis of quantitative measurements could independently support the thesis of ethnically diverse burials present at Machu Picchu [33].

## 5. Ceramic Chemical Analyses

The basis of ceramic archaeology in the Andes traditionally consists in classifying artefacts according to style and building relationships and chronologies between styles [34]. Over time, a certain style may become synonymous with the culture that manufactures it, however, this is increasingly confusing in the Inka period due to both the expansion of an imperial political force over the Andes, and the large-scale and long-range resettlement of peoples. This means that different styles are found all around the Andes, and it is unclear if they have been carried there or were made as imitations. Either way, the identification of a foreign style in a new place indicates the influence from the associated culture, either through cultural diffusion, contact or migrations. Recently, analytical chemistry methods have been employed to establish the makeup of the clays in different ceramics, which can distinguish between imitations and originals [35]. Furthermore, profiling the clays in a pot is an empirical quantitative way to classify them, whereas stylistic classification may be subjective, no matter how rigorous.

Using Instrumental Neutron Activation Analysis (INAA), one can identify the elements that make up a ceramic sample and in what proportion. These multivariate data then allow statistical identification of chemical compositional groups, which correlate to the specific recipe of different clays used to make the ceramic. The method was used recently to analyse ceramic samples from various Peruvian locations [35]. Of the 8 compositional groups identified in the study, all were represented among the 103 samples from Machu Picchu, and no one group represented > 50% of the samples [35]. This suggests multiple ceramic workshops were responsible for supplying Machu Picchu with its ceramic inventory. Similar evidence from other sites strongly suggests that each workshop produced ceramics for multiple locations, and therefore each workshop was not dedicated to the provision of a specific site [35]. Overall, this evidences a decentralised industry of ceramic production in the heartland of the Inka empire [35].

The INAA technique also allows differentiation between an “original” ceramic, made in one location and later transported elsewhere, and imitations, made with local clays but emulating the style from elsewhere. For example, samples from Pachacamac on the coast match the Cusco style but have chemical compositions that cluster more closely to other coastal and Lurin valley samples [35]. This suggests these artefacts were imitations made locally in the coast region, in the style of Cusco ceramics, and are indicative of some population having migrated to the coast from Cusco and using local resources to produce ceramics in their own styles. While the conclusions available from this kind of analysis are very limited, the technique could be invaluable if applied to ceramic artefacts in the “burial context” of a broader interdisciplinary characterisation of Inka-period burials in the Andes.

## 6. Genetics

Human genetic information is coded by three types of DNA: mitochondrial DNA (mtDNA), the sex chromosomes (X and Y) and autosomes. The mitochondrial DNA is located in subcellular energy-producing organelles called mitochondria and transmitted from mother to children. Its small size (~16.6 thousand base pairs), high copy number per cell and high mutation rate make it a convenient and cost-effective genetic marker to sequence from past and present individuals. Of the sex chromosomes, studies of past human genetic diversity mostly focus on the Male Specific region of chromosome Y (MSY). Transmitted from father to son, the chromosome Y is relatively large (~57 million base pairs) and present in a single copy per cell nucleus, which represents a challenge for sequencing and analysis, especially in ancient degraded samples. MtDNA and MSY are uniparental genetic systems (they are transmitted in a clonal fashion from mother to children and from father to sons, respectively) that allow tracing deep ancestry along maternal and paternal lineages, respectively. The autosomes are non-sex chromosomes (the bulk of the 6 billion base pairs of the human genomic DNA) present in pairs in the cell nucleus, with one copy inherited from the mother and one from the father. Ancestral genetic information recombines in autosomes at every generation, such that millions of variable genetic markers are reshuffled. This allows a high-resolution characterisation of demographic processes such as migration and genetic admixture between populations. In general, matrilineal and patrilineal genetic information can be used as a proxy for global settlement processes while autosomal DNA can inform both global and fine-scale regional demographic history. Genetics can also reveal the biological sex of a sample from the archaeological record [36,37,38], which can be a useful complement to physical anthropology methods. Sexing of individuals is particularly relevant for the analysis of sex ratio in *Aqllakuna*, *Yanakuna* and *Mitmaqkuna* burials.

Human population genetics studies utilise DNA from contemporary and/or past individuals. Contemporary DNA is sampled from living individuals and the computational analyses of sequenced DNA data allow inferences about the population history [39]. Ancient DNA (aDNA) is extracted from archaeological human remains and will have been significantly degraded as a result of decay processes that begin at the time of death. Ancient DNA is valuable for understanding past populations, as it directly captures genetic information at given time points and locations [39], therefore strengthening the population history inferred from present-day peoples [40].

Genetic studies of human populations compare patterns of genetic differences between individuals and/or populations to learn about population structure and origins [39]. In the Americas, this is more challenging than other world regions due to the relative lack of genetic diversity among Indigenous populations resulting from population bottlenecks during the entry into North America via Beringia [41,42,43,44,45] and the more recent population collapse following European contact [46,47]. Additionally, South America is extremely under sampled relative to other world regions, so studies with sample coverage to produce fine-scale demographic resolution are rare [48,49,50].

While archaeological studies of human activity in the Andes have been abundant [51], genetic studies are far scarcer but show that the region is characterised by high human mobility and genetic exchange [48,50,52,53,54,55]. High-resolution genome-wide aDNA data have only begun to be published in the last few years for pre-Columbian South America [46,56,57], with only two studies looking at the Inka period [55,58].

### 6.1. Genetic Continuity in the Chachapoyas Region Predates Inka

The Chachapoyas were a collection of ethnic groups particularly renowned for being subjected to Inka resettlement despite fierce resistance [18]. In a recent study, 119 present-day individuals from the Chachapoyas region were used to reconstruct demographic history by building a map of chromosome Y lineages and identifying geographic regions of shared Y-chromosome ancestry. Despite a bias because of the distribution of sampling locations, the genetic map did highlight a major exchange region in the Central Andes. It also identified high internal diversity among the Chachapoyas but differentiation relative to other Andean regions, including the major Central Andean exchange region [52]. This allows for either the possibility that people were moved to the Chachapoyas region during Inka times (allochthonous origin), or that the Chachapoyas groups descended from an amalgamation of more ancient Andean groups (autochthonous origin) [52]. The authors find an absence of Y-haplotype sharing with external populations, which would be expected had there been significant migration to the region during Inka times. While these results can only represent paternal population history, they strongly suggest that genetic continuity over the last 20 generations rather than migration [52]. This result does not definitively preclude some of the population from having been relocated away from Chachapoyas region, but it is evidence that enough of the population remained in the region to maintain genetic continuity and also that no significant number of *mitmaqkuna* were moved to the region.

### 6.2. Cosmopolitanism in Cusco and the Sacred Valley

This ancient DNA study is the most comprehensive reconstruction of Andean population history to date with 89 ancient individual genome-wide datasets, of which only 6 date to 1470–1602 CE (Late Horizon/Inka period) [55]. Results indicated long-range mobility and genetic heterogeneity at Cusco during the Inka period, interpreted as a signature of cosmopolitanism (i.e., multiple ethnicities living close together) [55]. This supports the previously proposed idea of Cusco as an “ethnic microcosm” at the centre of the empire [10]. Furthermore, significant heterogeneity in genetic diversity was identified in both Cusco and the Sacred Valley, where individuals appeared genetically close to populations from the north Peruvian coast, south Peruvian highlands and the Titicaca basin [55]. While it is important to note that not all geographical regions are covered in each archaeological time period in this dataset, the results are an early indication that further genetic study of mortuary populations at sites in the Inka heartland are needed for a better understanding of how the Inka constructed their capital and the multicultural demography of the region.

### 6.3. Tracing Mitmaqkuna in Northern Peru and Ecuador

A recent study of contemporary Indigenous populations from Peru and Ecuador aimed to investigate possible genetic connections between the Kañaris of Peru and the Cañaris in Ecuador, based on the toponym similarity and historical accounts mentioning a connection between the two [59]. Genetic affinities were compared at both the individual and population level using mitochondrial and chromosome Y DNA. Results showed no significant genetic connection between the Cañaris and Kañaris but there were several instances of likely *mitmaqkuna* individuals [59]. For example, some Ecuadorian individuals were closely related to the Kañaris and Inkawasi populations of northern Peru, suggesting that their ancestors were potential *mitmaqkuna* resettled during the Inka occupation [59]. Similarly, some individuals from the Cañaris (Ecuador) population were more closely related to individuals from northern Peru (Cajamarca) and southern Peru (Cusco, Chivay and Amantani), and could also be descended from *mitmaqkuna* [59]. Since these genetic results are from contemporary individuals, the instances of ectopic lineages indicate a probable migration, though researchers did not try to infer the timing of this event. Given the short time period in which Inka-mediated migrations could have occurred, it is indeed challenging to infer migration dates from modern DNA due to methodological limitations in this particular context. Further ancient DNA evidence would need to be collected from accurately dated archaeological burials to discriminate between Inka resettlement and earlier or later migrations.

## 7. Towards Interdisciplinary Studies to Better Understand the Inka Resettlement Policy

A recent paper integrated ancient DNA, archaeological, biochemical and historical evidence from 6 individuals from 2 cemeteries in the Chincha valley of southern Peru, which date to the Late Horizon (1400–1532 CE) and Colonial (1532–1825 CE) periods [58]. Whole-genome sequencing data revealed the Chincha individuals were genetically more related to ancient and present-day populations from the north Peru coast (several hundred kilometres away) than to populations from southern Peru [58]. This is strong evidence that the individuals or their ancestors had recently migrated to the Chincha valley. The genomic results were supported by archaeological findings of North and Central Coast style ceramics in the Chincha Valley and woven textiles of North Coast style [58]. This is particularly noteworthy given the imperial policy mandating *mitmaqkuna* individuals to maintain the production and wearing of their indigenous clothing style [14]. Strontium isotope abundance was also measured in bone tissues, but the results were inconclusive given the similar geology between all Peruvian coastal regions [58]. This study by Bongers and colleagues is the strongest yet combined set of empirical evidence for a case of the Inka state-enforced resettlement in pre-European contact Andes and it highlights the power of interdisciplinary research designs. Indeed, any one of the discipline-specific datasets in isolation would not lead to conclusions with the level of robustness achieved by collating all the interdisciplinary evidence together.

The genetic evidence collected so far gives preliminary support for impacts of the Inka empire on populations in the Andes [55], including some possible instances of relocated *mitmaqkuna* [58,59]. However, the low number of individuals for which high-resolution genetic data are available is a critical limitation. Additionally, without the inclusion of burial context and other information, DNA alone cannot tell us why and when these individuals were buried in locations far away from their closest genetic relatives. Based on our review of the current literature, we call for further interdisciplinary research to better understand the reality of resettlement under Inka rule. Figure 3A outlines the types of evidence that would need to be combined from multiple disciplines to thoroughly study Inka resettlement policy. We suggest that an increased application of such interdisciplinary study designs would be essential for classifying the burial cohorts as local, *mitmaqkuna*, *aqllakuna* or *yanakuna* (Figure 3B). If the sampling of ancient individuals is increased across the region of Tawantinsuyu, it would be possible to map instances of migration during the Inka empire, and to statistically test resettlement estimates from ethnocultural accounts (such as D’Altroy’s estimate of a quarter to a third of the total population relocated [2,10]).

It may be highly informative to leverage the oral microbiome, which can be recovered from calcified dental plaque (dental calculus) frequently found on archaeological teeth and contains heritable microbes with rapid generation times and relatively fast mutation rates. This alternative genetic record can potentially track recent and rapid human migrations if and where the study of actual human remains is not possible for ethical or technical reasons [60,61,62]. Furthermore, dental calculus also preserves dietary proteins that may inform about cultural practices and/or geographic origin [63].

The archaeological study of funeral practices across Tawantinsuyu could provide some information regarding local versus Inka-type practices but strikingly, there is no systematic review (to our knowledge) of the diversity of burial traditions in the Inka sphere of influence. Nevertheless, we argue that the grave architecture or setting might not provide accurate or relevant details about a buried individual. For example, architecture could be constrained by regional practices or societal classes independently from an individual’s cultural or biological background. Conversely, grave goods such as textiles and ceramics might be more associated to the buried individual and therefore be more informative. We propose that it is important to focus on archaeological, bioanthropological and bioarchaeological evidence that can be collected directly from the individuals rather than from the more general archaeological context in order to infer knowledge relevant to the Inka resettlement policy.

In any case, it is evident that interdisciplinary data collection is essential to achieve the strongest characterisation of archaeological human remains as subjects of Inka-mediated resettlement.

## 8. Conclusions

According to the ethnohistorical record, the massive subjugation of peoples by the Inka elite in Tawantinsuyu brought about social policies of long-range mass relocation of between a quarter and a third of peoples in order to (a) prevent rebellion of allied indigenous groups, (b) promote an economically productive and coherent society and (c) integrate the mosaic of indigenous identities into a ubiquitous Inka identity. Still, almost everything that is known about the implementation of this policy is recorded in the inherently biased body of European colonial ethnohistorical documents.

As Andean scholarship has developed, empirical disciplines have emerged that use data to investigate the reality and/or processes of this resettlement policy. Morphometric analysis of different types of intentional cranial deformation can be useful in the Andes to identify migrants given both the widespread and variable nature of the practice and due to its being acquired early into childhood. Chemical analyses of ceramic artefacts may also play a role in identifying imitation ceramics as opposed to imported originals. Additionally, stable isotopic analysis of teeth and bone material has allowed researchers to infer if buried individuals had migrated during their lifetimes. Though we caution not to over interpret the data into the presumed historical context. The isotopic studies mentioned have used historical knowledge of the migration policy as the basis to investigate demography at individual sites. However, many details about migration under Inka rule and sacrificial practice remain unknown, so the inferred demographic information at various sites must also be used to robustly quantify and characterise the migration policy.

Most recently the field of paleogenetics, which relies on the study of ancient DNA, has emerged. This has enabled the empirical examination of genetic relationships between individuals and/or populations of interest in the Andes, thereby providing the tools to establish which population a particular buried individual is most closely related to and therefore the population they may have migrated from. The genetic studies discussed herein are those of relevance to investigating the Inka resettlement policy and their small number is a further reminder that the Andes region is still massively understudied. In particular, current sampling of ancient individuals is too sparse to allow any robust quantification of the scale and magnitude to which resettlement occurred throughout the empire.

Now, with an abundance of available tools from different disciplines, researchers have the ability to combine population of origin (genomics), migration history through life (isotopic analysis) and class and identity (archaeological burial context) in order to characterise archaeological human remains within the framework of Inka resettlement identities given by ethnohistorical documents. Each discipline offers evidence towards a better understanding of the resettlement policy, but it is unlikely that any single discipline could accurately or adequately reach a robust outcome. Integration of all empirical lines of evidence must inform the anthropological characterisation of Inka resettlement policy. We propose that by applying an interdisciplinary approach throughout the Andes, the true processes of the Inka resettlement policy can accurately be described.

## Figures and Tables

**Figure 1 genes-12-00215-f001:**
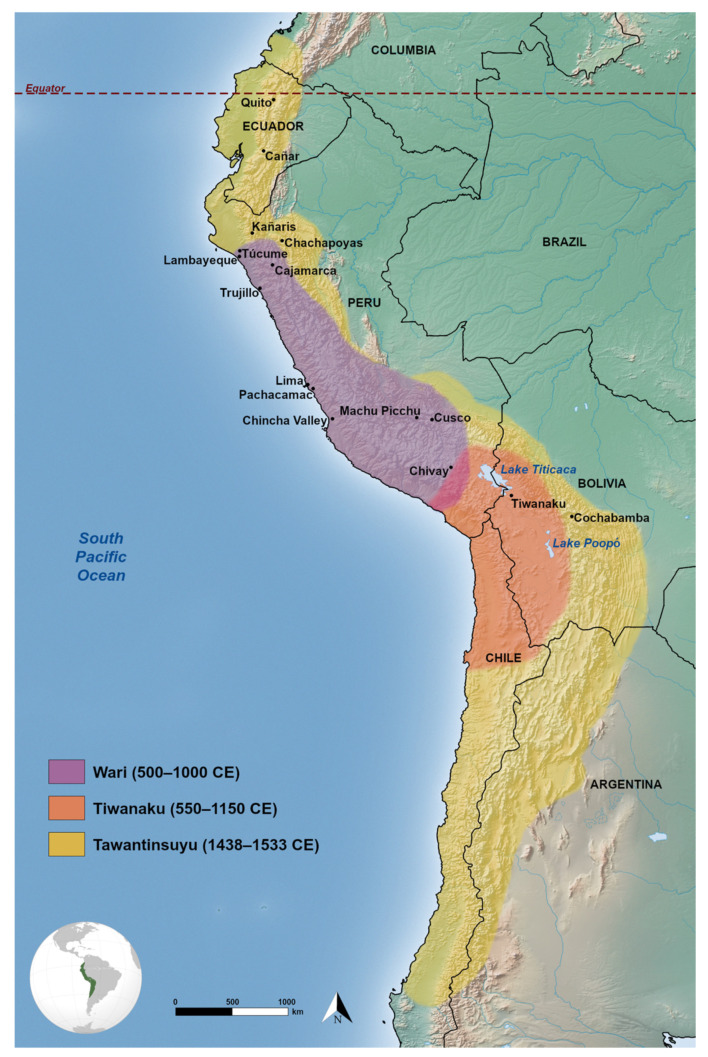
Political regions in the Andes. The Tiwanaku sphere (orange) and Wari sphere (purple) are overlayed with the maximal range of Tawantinsuyu (yellow). Locations mentioned in the text are also shown. Globe inset shows the location of Tawantinsuyu in the American continents. Made with Natural Earth.

**Figure 2 genes-12-00215-f002:**
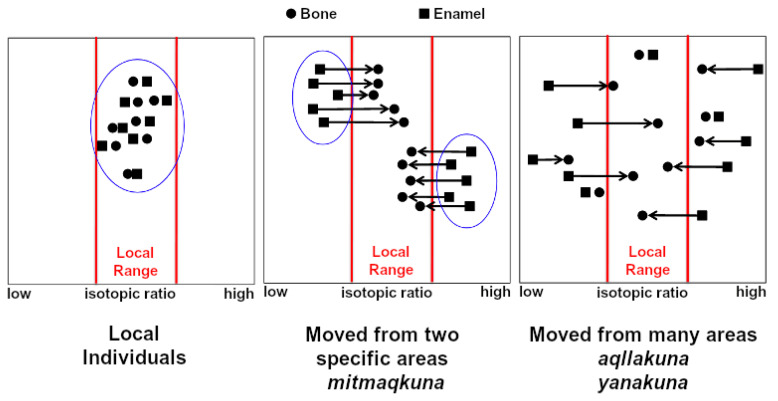
Schematic comparison of isotopic patterns of residential mobility for local individuals, *mitmaqkuna* or *aqllakuna*/*yanakuna*. Vertical axis separates samples in space. Modified with permission from [13].

**Figure 3 genes-12-00215-f003:**
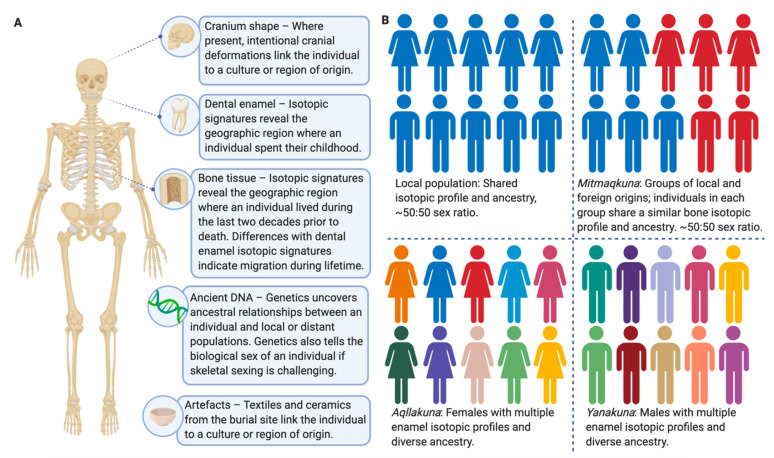
Schematic summary of evidence that could be combined from multiple disciplines to comprehensively address the Inka resettlement policy. (**A**) Evidence that could potentially be obtained from a buried individual. We strongly advocate merging multiple lines of evidence into an interdisciplinary framework. (**B**) Possible demographies of a hypothetical buried population. Individuals from a local population with no migration cluster together genetically, show no indication of migration during their lifetime, and a ~50:50 sex ratio is observed. *Mitmaqkuna* would be indicated by a significant proportion of individuals with isotopic evidence of during-life migration and shared ancestry distinct from the local population, and a ~50:50 sex ratio is expected. Groups of buried females or males with multiple isotopic profiles from enamel (early life) and diverse ancestry are likely to be *aqllakuna* or *yanakuna*, respectively. Created with BioRender.com (accessed on 2 February 2021).

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
