# Peer review of "A Multidisciplinary Review of the Inka Imperial Resettlement Policy and Implications for Future Investigations"

_genes, 2021, doi:10.3390/genes12020215_

Round 1
Reviewer 1 Report
The article entitled “An interdisciplinary review of the Inka imperial resettlement policy and implications for future investigations” by Davidson et al. presents a review of three main fields of research (history, archaeology, and genetics) focusing on reconstructing the history of population movements during the Inka Empire era.
While I think this relatively short review may be of interest to readers, I have several comments that would need to be addressed by the authors to improve the paper allowing for its publication.
- No specific section deals with funeral practices during the Inka Empire era from anthropological and archaeological perspectives. This is a problem as numerous aspects of the paper relies on the study of biological and cultural material excavated from burial sites. However, the authors do not provide basic knowledge about these important aspects, further making it difficult to discuss results obtained from anthropology, archaeology, and paleogenomic studies.
- While different research fields are explored by the authors, there is, I feel, a lack of discussion contrasting the results, except from specific subsections of the paper. These are reviewing the discussion made by the authors themselves in the example papers the authors dwell on, but a more general and systematic comparison of the different sources of knowledge will provide further depth to the paper.
As a non-exhaustive example, Appendix A is not discussed in terms of genetics categories (gene genealogies vastly differ from genealogies).
- I would argue that the authors propose pluri or multidisciplinarity approaches rather than interdisciplinary per se. This is further illustrated by the fact that they review separately the outcome of the different fields of research in separate subsections. Indeed, there is a difference between juxtaposing several research fields which can, as the authors emphasize throughout their paper, fill numerous gaps in the multifaceted history of human populations, and true interdisciplinarity, where different research paradigms are revisited to dialogue and influence one another.
- Finally, while the English is most often correct, I think, language is, I think, often difficult to read and a bit confusing at places, and would need thorough revision and clarification (see below for an unexhaustive list of punctuation, tense concordance, grammatical etc. issues).
- Overall, I feel that there is relatively few genetics in this paper, as there is few methodological aspects and discussion explored by the authors (as compared to isotopic studies and expectations for instance). I agree with the authors that the Inka Empire topic as been, relatively, seldom explored hence providing a limited amount of material to review.
Detailed chronological comments:
Appendix A: it is not clear why the authors need an Appendix in such a short review. I believe that the authors could conduct a thorough anthropological section about categorization issues and how they could influence recruitment and sampling in genetics, paleogenetics and archaeological studies. In any case, while of interest in itself, this appendix should be integrated to the paper, somehow, or abandoned I think.
L20: “resettlement policy” perhaps ? Sentence needs to be clarified
L21: “investigate” I believe here
L23: “, and”. Overall, there are issues with punctations throughout the whole paper that often result in misleading or unclear phrasing. I have noted some of them below but probably not all. Please, verify accordingly everywhere.
L24: “particular question”: it is not clear what the question in fact is. Authors could try to make it more explicit in the abstract. Similarly, in the Introduction, the set of questions the authors wish to address in this review should be made clearer.
L37: “, so” not clear. The authors often use this phrasing that could easily be modified to be a bit more elegant, I think.
L38: “tiered decimal hierarchy system”: please, explain for the non-specialized audience
L39: “existing” ?
L53: “through the lens of foreign scholarship and colonial loss of individual Indigenous identities.”: please clarify
L54: “soon”: please if possible, be more precise here as the duration of the Inka empire is short and overlaps with the Spanish colonization era
L58: “subject group” ? “Subjugated group” perhaps better ?
L62: “significant fraction”: could the authors attempt to be more precise here with some rough % ? I know that a whole section is dedicated to this below, but here, it feels a bit short, still.
Fig1: A latitude - longitude grid is needed, the north should be indicated, Pacific ocean or other sea geographical identifiers could be envisioned, and rivers and lakes could be shown in blue on the map
Authors could map the various political regions they talk about in the Inka Empire. The map feels not as informative as it could be.
L80: “would” check tenses
L85: embellishments
L86-91: I agree fully, but this is a bit trivial or even somehow unfair: scientific method in the early 16th century barely existed in a first place. While I agree with the authors, caution is perhaps a bit over emphasized here, and could be shortened.
L102: “Though”, there is no contradiction per se in the sentence: it was not slavery, and it has been compared to medieval European serf systems.
L109: “were” in italic
L117: “an order of magnitude” should probably not be used to measure “a level of difficulty” in a scientific article: it applies, mathematically, to mathematical objects, not “difficulties”.
L125-127: Important sentence, very hard to read. Please consider clarifying
L131: “Debate can be had” – “debatable”: Problem in the sentence. More generally, here is an example of a passive form that could be avoided for clarification. Check these throughout the text
L144: “became” instead of “was now”
L158: “now” should perhaps be “not” ?
L180: “, and”
L181-182: connection with “Inka period burials” is unclear, please rephrase.
L183: “will” inappropriate tense
L191: replace “by” with “with”. “A river catchment” is a bit unclear, reshape the sentence for clarification
L194: “, and”
L199: “entire mortuary populations” is unclear. Entire buried populations ?
Major comment: A paragraph concerning burial practices, sex-biased recruitments, social stratification, and/or practices related to immigrant status should be added for clarifying the isotopic-assumptions and hypotheses tested.
L232-233: it is not clear how “variation” in itself is interesting. The sentence is not very clear.
L235: Perhaps “broadly” with “extensively” ? check word order: “broadly documented” instead of “documented broadly” here.
L237: Replace “this is done…” with “ICDs are performed at a young age”
L240: Replace “intrigue” with “attention”
L244: “practise” should perhaps be “practice”
L247: “retainer population” is unclear
L257: “revolves around” is too unprecise the characterize the fundamental archaeology paradigms
L261-262: Tense concordance issues with previous sentence and within the sentence
L265: shouldn’t “analytical chemical methods” be “analytical chemistry methods” ?
L267: Multivariate data or multivariate analysis ?
L274: “no one group in majority” is unclear (absolute majority ?), actual proportions might be more informative here
L281: “imitations, made with local clays but imitating”, somehow unfortunate phrasing
L284: “would be” check tense concordance with surrounding sentences and within the sentence
L290: I believe authors are only talking about human population genetics studies, rather than human genetics studies as a whole.
L293: “will be” tense issues
L296: it is not enhancing modern data analysis: it tells a possibly different history as modern population genetics patterns may not be inherited from the observe ancient ones, for instance in the case of population replacements… It complements might be better, but still not completely accurate.
L299: the method itself is not limited by these historical peopling processes: it is this history that results in the complex patterns observed today, but the method only does what it is meant to do: reconstruct the forces that gave birth, over time, to the observed patterns
L301-303: Agreed, this is the major knowledge limitation: few samples and few investigations overall
L310: “collection ethnic group” ? It is unclear what this means
L321-323: “some” italicized
The sentence is a bit too definitive: Y chromosome lineages only reflect a very limited part of the genealogies: that of only a single male among all males in the genetic genealogy, for each sample.
Furthermore, other demographic processes such as philopatry and sex-specific transmission of reproductive success, may also play a role in explaining the observed patterns, not only migration.
L325: “This aDNA study…” Which one ? The paragraph above is about modern DNA if I am not mistaking
L349-350: this is not correct, authors did not try to infer migration/admixture times, but numerous methods exist to do so in population genetics inferences (maximum-likelihood, ABC, etc. model and simulation based for instance). Perhaps the type and amount of genetic data the authors used only allow limited such inference, but it is not the fact that they are contemporaneous data, that limits the inference. Otherwise, population geneticists would not use modern genetic data to make population history inferences.
For this whole paragraph, authors should describe better the individual inclusion criteria used in these studies. Disentangling recent migrations from historical ones during the Inka empire, may primarily result from who exactly is studied in these articles.
L366: The study… Tense concordance issues throughout the paragraph
L383: if it is evident, it is perhaps not necessary to include the sentence here…
L394: no need to further write ICD acronym here
L396-397: contradicts the above review: we do not know if the ceramics are produced by migrants or whether only the technics and materials have moved.
L407: population genetics or paleogenomics are not “mechanisms”
Reviewer 2 Report
This is a very well written manuscript that covers a great amount of scholarship in a concise and effective manner. The authors are quite convincing in their argument that a mult-disciplinary approach is needed to learn more about the relatively unknown elements of the Inka empire, which mainly stem from biased or potentially inaccurate records from Spanish colonizers. I very much appreciated the caveats presented for each method of analysis, including the limitations of isotope and cranial studies. I only have a few suggestions: 1. The authors mention different papers that have utilized uniparental markers to make assessments of migration. I think it's important to add the limitations of these types of studies to make such claims. 2. I think it would be interesting to describe a future study design that could answer the migration questions in a comprehensive manner. What type of sites and samples would be needed for such a study? 3. The authors mention whole genome and genome-wide studies performed in the region. I think it would be worth while to mention the methods used to assess migrations that would not be available to studies using uniparental markers. 4. Lastly, do the authors see any role for the use of metagenomics derived from dental calculus that could help solve the riddle of complex migrations in the Andes?
